# Surveillance for TB drug resistance using routine rapid diagnostic testing data: Methodological development and application in Brazil

**Sarah E. Baum** [1]*, **Daniele M. Pelissari**[2], **Fernanda Dockhorn Costa**[2], **Luiza O. Harada** [2], **Mauro Sanchez**[3], **Patricia Bartholomay** [2], **Ted Cohen**[4], **Marcia C. Castro**[1], **Nicolas A. Menzies**[1]

**1** Department of Global Health and Population, Harvard T.H. Chan School of Public Health, Boston, Massachusetts, United States of America, **2** Health and Environment Surveillance Secretariat, Ministry of Health, Brasília, Brazil, **3** Department of Public Health, University of Brasília, Brasília, Brazil, **4** Department of Epidemiology of Microbial Diseases, Yale School of Public Health, New Haven, Connecticut, United States of America

* sbaum@g.harvard.edu

**Data Availability Statement:** Associated code is available on a public GitHub repository at https://github.com/sbaum95/Brazil_TB_Rif. Secondary

## Abstract

Effectively responding to drug-resistant tuberculosis (TB) requires accurate and timely information on resistance levels and trends. In contexts where use of drug susceptibility testing has not been universal (i.e. not all patients are offered testing), surveillance for rifampicin-resistance—one of the core drugs in the TB treatment regimen—has relied on resource-intensive and infrequent nationally-representative prevalence surveys. The expanded availability of rapid diagnostic tests (RDTs) over the past decade has increased testing coverage in many settings. However, RDT data collected in the course of routine (but not universal) use may provide biased estimates of resistance if the subset of patients receiving RDTs is not representative of the overall cohort. Here, we developed a method that attempts to correct for non-random use of RDT testing in the context of routine TB diagnosis to recover unbiased estimates of resistance among new and previously treated TB cases. Specifically, we employed statistical corrections to model rifampicin resistance among TB notifications with observed Xpert MTB/RIF (a WHO-recommended RDT) results using a hierarchical generalized additive regression model, and then used model output to impute results for untested individuals. We applied this model to 2017–2023 case-level data on over 800,000 patients from Brazil. Modeled estimates of the prevalence of rifampicin resistance were substantially higher than naïve estimates, with estimated prevalence ranging between 28–44% higher for new cases and 2–17% higher for previously treated cases. Our estimates of RR-TB incidence were estimated with narrower uncertainty intervals relative to WHO estimates for the same time period, and were robust to alternative model specifications. Our approach provides a generalizable method to leverage routine RDT data to derive timely estimates of RR-TB prevalence among notified TB cases in settings where testing for TB drug resistance is not universal.

and de-identified data from SINAN are available online on the official webpage of the Ministry of Health (MoH) of Brazil. These data can be accessed through the link: https://datasus.saude.gov.br/transferencia-de-arquivos/.

**Funding:** S.E.B was supported by a T32 National Research Service Award (T32 A1007535-23) from the United States National Institute of Allergy and Infectious Diseases (https://researchtraining.nih.gov/programs/training-grants/T32-a). The funders had no role in the study design, data collection and analysis, decision to publish, or preparation of this manuscript.

**Competing interests:** The authors have declared that no competing interests exist.

## Author summary

While data on drug-resistant tuberculosis (DR-TB) may be routinely collected by National TB Control Programs using rapid diagnostic tests (RDTs), these data streams may not be fully utilized for DR-TB surveillance where low testing coverage may bias inferences due to systematic differences in RDT access. Here, we develop a method to correct for potential biases in routine RDT data to estimate trends in the prevalence of TB drug resistance among notified TB cases. Applying this approach to recent national case-level data from Brazil, we find that modeled estimates were higher than naïve estimates, and with narrower uncertainty intervals compared to estimates produced by the World Health Organization. We highlight the value of this approach to settings where testing coverage is low or variable, as well as settings where coverage may surpass existing coverage thresholds, but that could nonetheless benefit from additional statistical correction.

## Introduction

In 2022, 410,000 individuals were estimated to have developed tuberculosis (TB) that is resistant to rifampicin—a key component in the WHO-recommended first-line treatment regimen [1]. Even with prompt detection and appropriate treatment, drug-resistant TB is more challenging to treat than drug-susceptible TB [2]. As a result, individuals with drug-resistant disease experience worse health outcomes, such as higher rates of treatment failure and mortality [3,4]. In settings where routine use of drug susceptibility testing has not been universal, surveillance for TB drug resistance has historically relied on nationally representative prevalence surveys, which can be resource-intensive and are therefore conducted infrequently [5].

Monitoring resistance to rifampicin is of particular importance for drug resistance surveillance [6]. Rifampicin resistance often occurs in the presence of resistance to isoniazid [7]. Together these drugs form the backbone of the first-line regimen, with resistance to both defining multi-drug resistant TB (MDR-TB). Innovations in rapid diagnostic tests (RDTs), such as Xpert MTB/RIF, have enabled point-of-care detection of both tuberculosis and rifampicin resistance [8]. RDTs have progressively been integrated into routine TB care since arriving on the market [9,10]. As of 2023, nearly half of all cases newly diagnosed with TB were initially tested with a WHO-recommended RDT [1], and global TB strategy targets assume that RDT coverage will continue to increase [11].

Increasing use of RDTs presents new opportunities for rifampicin-resistant TB (RR-TB) surveillance, as these tests provide information on the presence of resistance amongst tested individuals [12]. However, these data can yield biased estimates if RDT coverage is not universal, or if there are systematic differences in access to testing. Systematic differences in testing may arise, for instance, if there is geographic variation in access to RDTs (e.g., urban or rural locations, proximity to health facilities or laboratories that perform testing) or if patients with certain observable characteristics are more likely to be offered testing. The potential for bias is likely to be more pronounced when testing coverage is lower. Predicting the direction of the potential bias is not obvious at the outset; if RDT use is more common in regions of a country, health facilities, or patient groups with a higher (lower) prevalence of resistance, then the simple average of RDT rifampicin resistance results will overestimate (underestimate) true resistance levels.

Ensuring that routinely collected data are sufficiently complete before using them for resistance surveillance can minimize potential biases induced by non-random testing. For this

reason, the WHO Global TB Programme does not use these routinely collected data for resistance surveillance unless at least 80% of bacteriologically confirmed pulmonary TB cases are routinely tested for drug susceptibility. As a result, however, estimates of TB drug resistance in countries where routinely-collected data are not sufficiently complete may still rely on potentially outdated prevalence surveys [1]. Thirty-three countries did not meet this threshold in 2022. Based on WHO estimates of TB incidence, countries that rely exclusively on national surveys to estimate TB drug resistance prevalence among new cases accounted for almost 25.7% of the estimated annual number of global incident TB cases, while 105 countries that exclusively rely on continuous surveillance accounted for 3.4% [13].

Even when coverage is below WHO-suggested thresholds, RDT data could still deliver reliable and timely evidence on resistance trends if potential biases in coverage are identified and corrected using statistical methods. In this study we develop a novel surveillance approach that corrects for potential biases in routine RDT data from Xpert MTB/RIF tests in order to estimate levels and trends in the prevalence of rifampicin resistance among notified TB cases. We describe the conditions under which this approach will provide unbiased inference and demonstrate the approach using data from Brazil, which began implementing routine rapid diagnostic testing in 2014, but has not yet met WHO-suggested thresholds for continuous surveillance.

## Methods

### Study setting

As of 2023, Brazil was considered a high-burden country for TB but not for RR-TB by the WHO [1]. Brazil's most recent nationally-representative TB drug resistance survey occurred in 2006–2008, although these results are often regarded as preliminary and incomplete [14]. Brazil introduced Xpert MTB/RIF for routine TB diagnosis in 2014, and as of 2023, nearly 40% of notified TB cases had a conclusive Xpert resistance result.

### Data

Case-level TB data was extracted from Brazil's national Notifiable Disease Information System (SINAN). SINAN contains demographic and clinical data on all notified TB cases in the country. The sample was restricted to notified TB cases diagnosed between 2014–2023, which represents the period from when Xpert was initially rolled out through the most recent data available at the time of this study. We exclude individuals who were diagnosed with TB postmortem (n = 6,112; 0.7%), transferred to another facility (27,455; 2.9%), whose diagnosis type was unknown (4,274; 0.5%), and who were misdiagnosed with TB (10,100; 1.1%). We stratified these data according to whether an individual was a 'new' (no previous TB treatment, 738,922) or 'previously treated' (prior TB treatment, 153,202) case. Previously treated cases included individuals that had relapsed after previously being cured and individuals who defaulted from treatment for at least 30 days.

National and state population figures were based on Brazil's 2010 Census from the Brazilian Institute of Demography and Statistics (IBGE), which was the most recently available [15]. Coordinates for municipality latitude and longitude were sourced from IBGE's 2020 municipality shapefile [16].

### Inference framework

For each individual with notified TB disease $i$, from $i = 1,\ldots, N$, let $Y_i$ represent the presence of rifampicin resistance, equal to 1 for rifampicin-resistant disease and 0 otherwise. Let $T_i$ be a

binary indicator equal to 1 if the individual has a recorded Xpert resistance result, and 0 otherwise. $T_i = 0$ includes individuals who are not tested with Xpert, who are not positive for TB with Xpert despite receiving a TB diagnosis, who have an indeterminant rifampicin susceptibility result, and where the result was not recorded. As a result, $Y_i(0)$ is by definition unobserved.

We define the prevalence of rifampicin resistance among notified TB cases as E(Y):
$E[Y] = \frac{\sum Y_i}{N}$. If an individual's potential rifampicin resistance outcome is independent of whether or not they have a recorded resistance result ($\{Y_i(1),\ Y_i(0)\}_{i=1}^N \coprod T_i$), then the prevalence of rifampicin resistance in a given sample, $\bar{Y}_i$ [1], will be an unbiased estimate of positivity among all notified cases, $E[Y]$. This is violated, however, if the probability of receiving Xpert is not the same for all $i$, and if differences in testing are correlated with differences in rifampicin resistance. As a result, the subset of individuals with a conclusive Xpert resistance result will systematically differ from those who do not. The prevalence of rifampicin resistance among individuals with observed Xpert resistance results may not be equal to those who do not ($E[Y_i[1]|T=1] \neq E[Y_i[1]|T=0]$), which would lead to biased inferences about prevalence in the population of notified TB cases.

However, unbiased estimates of rifampicin resistance can be identified if the factors inducing biases in the Xpert testing data (i.e., those jointly correlated with Xpert coverage and rifampicin resistance levels) are known and observable. In this setting, testing can be considered random ($Y_i \coprod T_i|X_i$), conditional on a vector of covariates ($X_i$) that are jointly associated both with $T_i$ and $Y_i$. Assuming conditional independence holds such that $P(Y_i|T_i,X_i) = P(Y_i|X_i)$, we can recover prevalence of rifampicin resistance among all notified TB cases: $E[Y] = E[E[Y|X]] \equiv E[E[Y|T,X]]$.

Based on this inference framework we implemented a two-step statistical procedure to recover unbiased estimates of rifampicin resistance prevalence by: 1) modeling the probability of rifampicin resistance among individuals with recorded Xpert resistance results ($Pr(Y_i(1) = 1|X_i)$); and 2) using the modeled distribution, $f(Y|X)$, to impute the probability of rifampicin resistance for individuals without recorded results ($Y_i(0)$). This procedure will produce unbiased inferences if the set of predictors ($X_i$) that determine receipt of Xpert is known and observable in the data. Second, it requires there to be overlap in covariate distributions of individuals with and without recorded resistance results ($0 < Pr(T_i = 1|X_i = x) < 1$), such that $Y_i(0)$ can be imputed for all levels $X = x$.

## i) Model for RR-TB among observed Xpert results

We constructed a hierarchical generalized additive regression model to estimate whether an observed Xpert result is resistant, $Y_{itj}$, in state $j$ in quarter $t$. The probability of an RR-TB positive test result ($Pr(Y_{ijt} = 1) = \pi_{ijt}$) was modeled as:

$$Y_{ijt} \sim Bernoulli(\pi_{ijt}) \tag{1}$$

$$\text{logit}(\pi_{ijt}) = \zeta_j + f_j\left(\text{quarter}_{ij}\right) + f\left(\text{latitude}_i, \text{longitude}_i\right) + X_i\beta \tag{2}$$

where ($\pi_{ijt}$) was a function of random intercept for state ($\zeta_j$), a state-specific smooth time trend ($f(\text{quarter}_{ij})$), time invariant spatial variation based on the centroid of individual's municipality of residence ($f(\text{latitude}_i, \text{longitude}_i)$), and a set of time-invariant individual covariates ($X_i\beta$) potentially associated with being tested as well as the risk of acquiring or developing RR-TB. The set of covariates–age, sex, HIV status, and level of care offered by the diagnosing health facility–were identified based on the set of covariates available in SINAN and through

discussion with Brazil's National TB Control Program. Thin-plate regression splines were used to reduce variance in state-level time trends ($f(\text{quarter}_{ij})$) due to sampling uncertainty. Similarly, a two dimensional thin-plate regression spline was used to allow geographic differences within states ($f(\text{latitude}_i, \text{longitude}_i)$).

The model was fit to the set of recorded Xpert resistance results, defined as being either rifampicin resistant or susceptible. This excluded observations in which Xpert was not performed and where Xpert was performed, but where TB was not detectable, the result was inconclusive, or the result was not recorded. Separate models were fit for new and previously treated patients. Our primary model was fit to 2017–2023 to exclude two periods of relatively low coverage when Xpert was rolled out in 2014–2015 and during a period of cartridge stocks outs in 2016.

### ii) Imputation of missing Xpert test results

Among the subset of new and previously treated individuals with a missing Xpert result (i.e., not tested, TB undetectable, or an indeterminant susceptibility result), we used the fitted models from part i) to impute the probability of a rifampicin resistant test result. These revised data (including both observed and imputed values) were then used to produce estimates of rifampicin resistance among all TB notifications.

### Statistical analysis

Point estimates of rifampicin-resistance prevalence among notified TB cases for each area $j$ at time $t$ ($\mu_{jt}$) were calculated by summing non-missing resistance results and the imputed probabilities, divided by total TB notifications ($n_{jt}$) in area $j$ at time $t$ [1].

$$\mu_{jt} = \frac{\sum_{ijt}^{T_i=1} Y_{ijt} + \sum_{i}^{T_i=0} \pi_{ijt}}{n_{jt}}$$

Total rifampicin-resistant notifications per 100,000 person-years in area $j$ at quarter $t$ ($\Psi_{jt}$) were calculated by dividing the modeled number of cases by the population ($N_{jt}$) in area $j$ at quarter $t$ [2].

$$\Psi_{jt} = \frac{4 * \left\{ \sum_{ijt}^{T_i=1} Y_{ijt} + \sum_{ijt}^{T_i=0} \pi_{ijt} \right\}}{N_{jt}} * 100,000$$

We compared these estimates to those produced by a naïve approach assuming Xpert results were missing at random. The prevalence of rifampicin resistance was calculated as the share of conclusive Xpert resistance results that were resistant. Total RR-TB notifications per 100,000 person-years were calculated by dividing the number of resistant Xpert tests scaled by the fraction tested by the national or state-level population.

Finally, to account for under-detection, we calculated total rifampicin-resistance incidence by adjusting estimates according to Brazil's annual case detection rate (CDR), as estimated by WHO under the assumption that the probability of resistance was the same across notified and non-notified TB cases [17]. Since CDR estimates were only available through 2022, we applied 2022 estimates to 2023.

We constructed 95% uncertainty intervals using a simulation approach. First, we generated 1,000 samples from the uncertainty distribution of the coefficients of the fitted regression models. We summed the recorded resistant Xpert results and the simulated probabilities imputed for missing test results to obtain the modeled number of RR-TB cases for each

simulation. We then calculated the 2.5th and 97.5th percentiles from the simulated cases counts, aggregating at the national- and state-level by quarter or year. All analyses were performed in R (2022.12.0+353) using the mgcv package (1.9–0) [18].

## Comparison to WHO annual estimates

Modeled estimates were compared to the WHO Global TB Program annual estimates of the proportion of notified TB cases with rifampicin resistance by case type and total rifampicin resistance incidence for Brazil. Since WHO estimates were only available through 2022, 2022 estimates were carried over for comparison in 2023 [13].

## Alternative model specifications

We compared 2017–2023 model results to estimates for the full period since Xpert implementation (2014–2023) to understand how the model performed with low levels of Xpert coverage among notified TB cases. We also fit two alternative model specifications to explore whether estimates were robust to these design choices. First, we refit the regression model with additional patient characteristics (e.g., educational attainment, race, and indicators for diabetes, illicit drug consumption, tobacco consumption, alcohol consumption, housing status, incarceration status, and immigration status) included as fixed effects. Second, we refit the regression model with separate interactions between the patient characteristics included in the main model (e.g., age, sex, HIV status, and health unit) and the smooth time trend, to explore whether the type of patients being tested with Xpert was static overtime, as assumed in the main model.

## Human subjects protections

The Institutional Review Board of the Harvard T.H. Chan School of Public Health determined that this study did not qualify as human subjects research (Protocol Number: IRB24-0009).

## Results

### Xpert coverage and observed RR-TB trends

Except during a period of stock outs in 2016, national coverage of Xpert among notified TB cases in Brazil increased in all years between 2014–2023, from 2.8% in 2014 to 41.0% of cases in 2023 (Fig 1A). Among those tested over the period, 1.6% of cases had an inconclusive result and 4.9% had a non-detectable TB result. Xpert coverage among previously treated cases was consistently about 4–5 percentage points higher than new cases, from 2015 onwards. During the period of lowest Xpert coverage between 2014–2016, naïve estimates of rifampicin-resistance prevalence was, on average, 5.9% and 10.6% among new and previously treated cases, respectively. Between 2017–2023 (when Xpert coverage was substantially higher), 4,553 new (2.7%) and 2,380 previously treated (5.8%) cases had a rifampicin resistant Xpert test result.

Table 1 presents descriptive statistics on all notified TB cases. The majority of cases were male, aged 20–50, and sought care at a low complexity health facility. S1 Table presents evidence of systematic differences in Xpert access and rifampicin resistance levels according to predictors included in the main model. Among certain factors, such as a sex and age, greater Xpert access is inversely associated with lower rifampicin resistance.

### Estimated national trends in RR-TB prevalence

Fig 1B compares estimates of rifampicin resistance produced by our proposed approach compared to naïve estimates calculated by averaging recorded Xpert resistance results, relative to

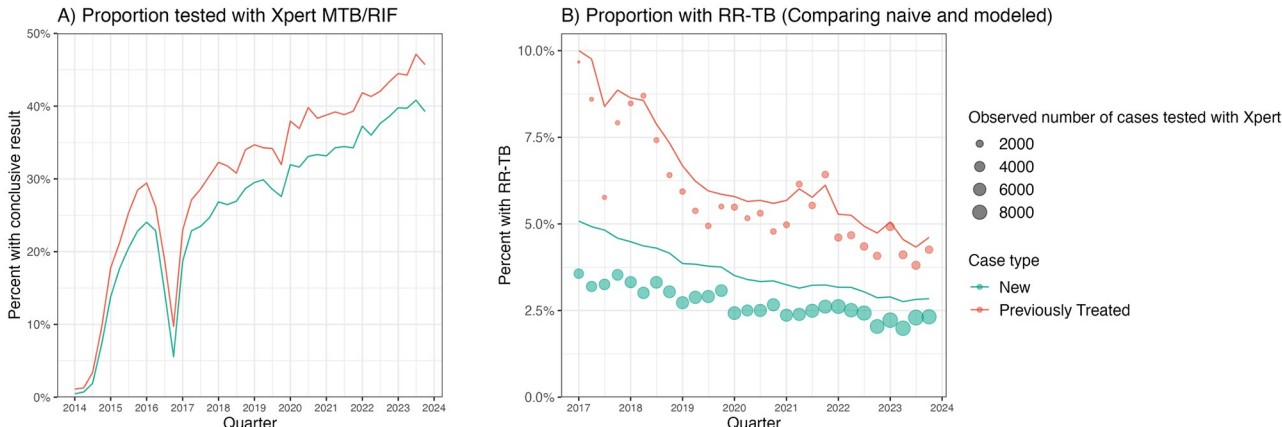

**Fig 1. Trends in Xpert test coverage and modeled prevalence of rifampicin resistance among notified TB cases.** (A) The proportion of new and previously treated TB cases tested with Xpert between 2014–2023 with a conclusive rifampicin resistance result, defined as being either susceptible (RR-TB negative) or resistant (RR-TB positive). This excludes observations where patients were tested with Xpert, but their result was labeled as indeterminant, not recorded, or was not positive for TB with Xpert despite receiving a TB diagnosis. (B) Lines reflect the modeled prevalence of rifampicin resistance among all notified TB cases for a given quarter by case type for 2017–2023. Points reflect the prevalence of rifampicin resistance calculated using the naïve approach, defined as the share of all conclusive Xpert results with rifampicin resistance. Point size indicates the number of notified TB cases with a recorded Xpert rifampicin test result.

the changes in Xpert coverage among notified TB cases over time. Modeled estimates of the prevalence of rifampicin resistance were higher than the naïve estimates for new and previously treated cases. This indicates that the coverage of Xpert testing differed systematically across notified TB cases and was negatively correlated with the probability of rifampicin resistance. The extent to which the naïve estimates underestimated RR-TB burden was more apparent for new cases than for previously treated cases. Fig 2 quantifies this difference as the ratio of modeled to observed RR-TB prevalence per 100,000 person-years. Between 2017–2023, the average bias was higher for new cases (34%; 95% uncertainty interval (UI): 28%, 43%) compared to previously treated cases (10%; 95% UI: 3%, 20%). Bias declined over the study period alongside increases in testing coverage, with greater declines for new compared to previously treated cases.

In 2023, we estimated the annual number of notified TB cases with rifampicin resistance to be 1.8 per 100,000 person-years (95% UI: 1.7, 1.9) (Fig 3). When accounting for under-detection of TB, this increased to 2.1 cases per 100,000. We estimated prevalence of rifampicin resistance among notified TB cases to be 2.8% for new cases (95% UI: 2.7%, 3.0%) and 4.6% for previously treated cases (95% UI: 4.5%, 5%). Both new and previously-treated individuals experienced declines in the number of rifampicin resistant cases between 2017 and 2023 –a 31.8% reduction for new cases and a 23.1% reduction for previously treated cases.

## Estimated state-level RR-TB trends

Fig 4 presents the prevalence of rifampicin resistance by state in 2023. For new cases, this ranged from 8.4% in Mato Grosso (95% UI: 6.3%, 12.0%) to 0.3% in Roraima (95% UI: 0.2%, 0.6%), and several states with the highest positivity were located in the Northeast region. For previously treated cases, prevalence of rifampicin resistance ranged from 14.3% in Maranhão (95% UI: 11.7%, 18.5%) to 0.2% in Sergipe (95% UI: 0.1%, 0.9%). Fig 5 presents trends in rifampicin-resistant cases per 100,000 person-years for states with the highest modeled prevalence in 2023. While modeled trends in the national prevalence of resistance declined between

**Table 1. Descriptive characteristics of notified TB cases (2017–2023).**

| Variable | New | | | Previously Treated | | |
|---|---|---|---|---|---|---|
| | All TB notifications (N = 458,101) | Recorded Xpert resistance result (N = 155,820) | Missing Xpert resistance result (N = 302,281) | All TB notifications (N = 100,370) | Recorded Xpert resistance result (N = 39,614) | Missing Xpert resistance result (N = 60,756) |
| **TB notifications with recorded Xpert resistance result** | 0.34 | - | - | 0.39 | - | - |
| **Recorded Xpert result with resistance indicated** | - | 0.03 | - | - | 0.05 | - |
| **Male**[a] | 0.69 | 0.72 | 0.67 | 0.77 | 0.79 | 0.75 |
| **Age**[a] | 40.08 (17.62) | 38.88 (16.38) | 40.70 (18.20) | 39.78 (14.42) | 38.89 (13.44) | 40.36 (15.00) |
| **Race** | | | | | | |
| Brown | 0.50 | 0.52 | 0.50 | 0.50 | 0.51 | 0.50 |
| White | 0.28 | 0.26 | 0.29 | 0.24 | 0.24 | 0.24 |
| Black | 0.13 | 0.14 | 0.12 | 0.17 | 0.18 | 0.16 |
| Asian | 0.01 | 0.01 | 0.01 | 0.01 | 0.01 | 0.01 |
| Indigenous | 0.01 | 0.01 | 0.01 | 0.01 | 0.00 | 0.01 |
| **HIV Positive**[a] | 0.09 | 0.08 | 0.09 | 0.17 | 0.16 | 0.17 |
| **Has diabetes** | 0.09 | 0.09 | 0.09 | 0.06 | 0.06 | 0.06 |
| **Level of care at diagnosing health unit**[a] | | | | | | |
| Low complexity | 0.55 | 0.61 | 0.52 | 0.53 | 0.56 | 0.50 |
| Medium complexity | 0.26 | 0.25 | 0.26 | 0.30 | 0.30 | 0.30 |
| High complexity | 0.16 | 0.13 | 0.18 | 0.13 | 0.12 | 0.15 |
| Other | 0.03 | 0.02 | 0.04 | 0.04 | 0.02 | 0.05 |
| **Educational attainment** | | | | | | |
| No education | 0.04 | 0.03 | 0.05 | 0.04 | 0.03 | 0.04 |
| Some primary school | 0.30 | 0.33 | 0.29 | 0.39 | 0.41 | 0.38 |
| Completed 8th grade | 0.06 | 0.06 | 0.05 | 0.06 | 0.06 | 0.06 |
| Some secondary school | 0.13 | 0.15 | 0.12 | 0.12 | 0.14 | 0.11 |
| Completed secondary school | 0.11 | 0.11 | 0.11 | 0.07 | 0.07 | 0.07 |
| Some university | 0.03 | 0.03 | 0.03 | 0.01 | 0.01 | 0.01 |
| Completed university | 0.03 | 0.02 | 0.04 | 0.01 | 0.01 | 0.01 |
| **Immigrant** | 0.01 | 0.01 | 0.01 | 0.01 | 0.01 | 0.00 |
| **Homeless** | 0.03 | 0.04 | 0.02 | 0.10 | 0.13 | 0.08 |
| **Incarcerated** | 0.10 | 0.14 | 0.08 | 0.15 | 0.17 | 0.14 |
| **Uses tobacco** | 0.24 | 0.29 | 0.20 | 0.35 | 0.40 | 0.31 |
| **Uses alcohol** | 0.17 | 0.21 | 0.15 | 0.30 | 0.33 | 0.28 |
| **Uses illicit drugs** | 0.14 | 0.20 | 0.11 | 0.31 | 0.38 | 0.27 |

Means with standard deviations presented in parentheses. Sample is restricted to TB notifications for new and previously treated cases between 2017–2023. This excludes individuals diagnosed with TB post-mortem, transferred to another facility, whose diagnosis type was unknown, and those who were misdiagnosed with TB. "Recorded Xpert resistance result" refers to all notified TB cases with either a recorded resistant or susceptible Xpert resistance result. "Missing Xpert resistance result" includes individuals who were not tested with Xpert, who were not positive for TB with Xpert despite receiving a TB diagnosis, who had an indeterminant rifampicin resistance result, and where the result was not recorded. Level of the notifying health unit includes: Low complexity (first level of care), Medium complexity (second level of care, including TB referral networks), High complexity (tertiary level of care, including TB referral networks), and Other (e.g. lab, surveillance, clinics).
[a]Indicates covariates included in main model.

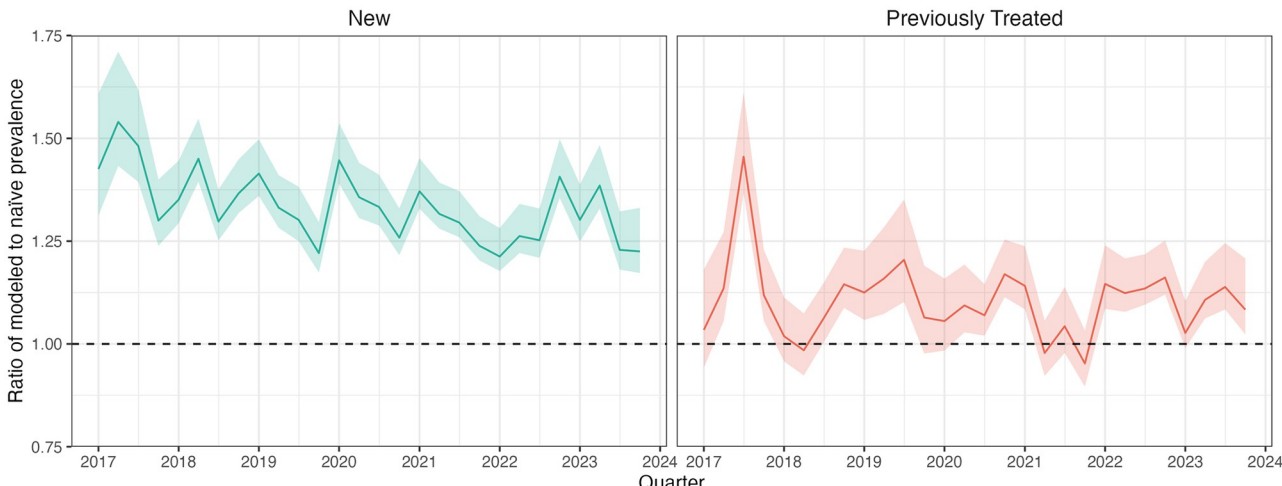

**Fig 2. Extent to which naive Xpert results underestimates the prevalence of rifampicin resistance among notified TB cases (2017–2023).** Each line reflects the bias in observed data as a function of the ratio of modeled to naïve prevalence among notified TB cases per 100,000 person-years. Prevalence of rifampicin resistance is calculated as either the modeled number of RR-TB cases (modeled) or the observed number of RR-TB cases scaled by the fraction with conclusive Xpert rifampicin results (naïve), divided by the national population in 2010. Numerators are calculated quarterly and are scaled to obtain person-years. This should be interpreted as how much higher estimates of rifampicin resistance prevalence using our approach are relative to naïve estimates. The dashed line at 1 indicates where there would be no bias between modeled and naïve estimates. 95% uncertainty intervals are shaded.

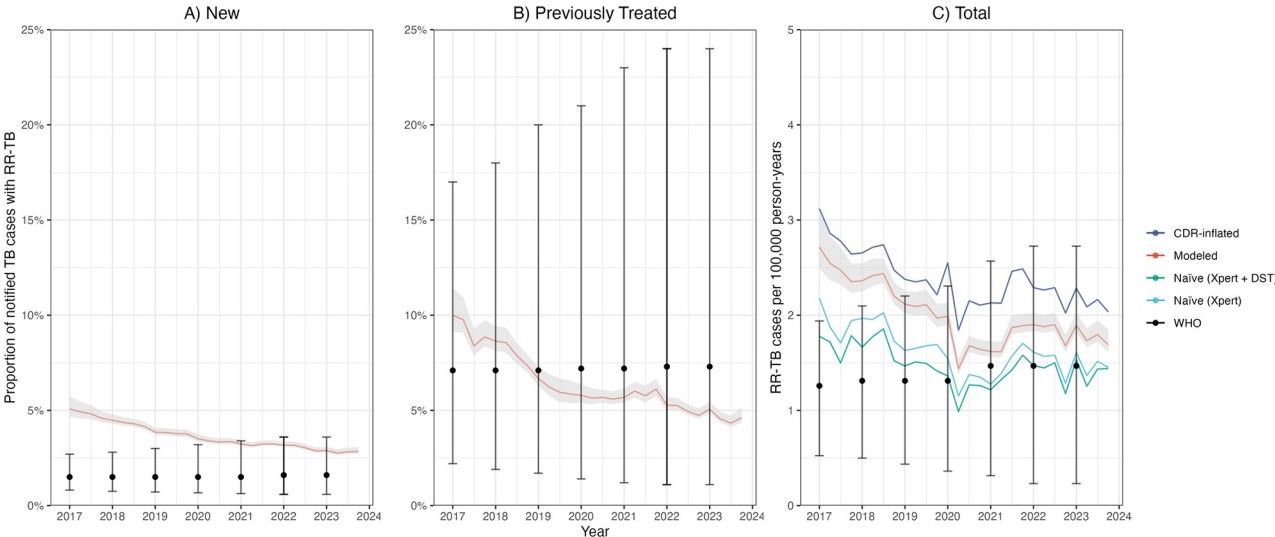

**Fig 3. Rifampicin resistance prevalence among notified TB cases and total incidence comparing modeled to WHO estimates (2017–2023).** (A) and (B) Modeled prevalence of rifampicin resistance by case type and 95% uncertainty intervals that are shaded. (C) Modeled total number of RR-TB cases among notified TB cases per 100,000 person-years and corresponding 95% uncertainty intervals ("Modeled"). It also presents total incidence after adjusting modeled estimates by Brazil's case detection rate (CDR) to account for underreporting of TB cases ("CDR-inflated") [16]. Naïve refers to the number of RR-TB cases calculated from Xpert MTB/RIF only ("Naïve (Xpert)") and from all DST results ("Naïve (Xpert + DST)") test results, scaled by the fraction of notified TB cases that were tested. DST results are included if resistance to at least rifampicin is indicated. All three panels are overlaid by the corresponding estimates and 95% uncertainty intervals from the World Health Organization's Global Tuberculosis [13]. Since CDR and RR-TB estimates from WHO are only available through 2022, 2023 estimates were carried over from 2022.

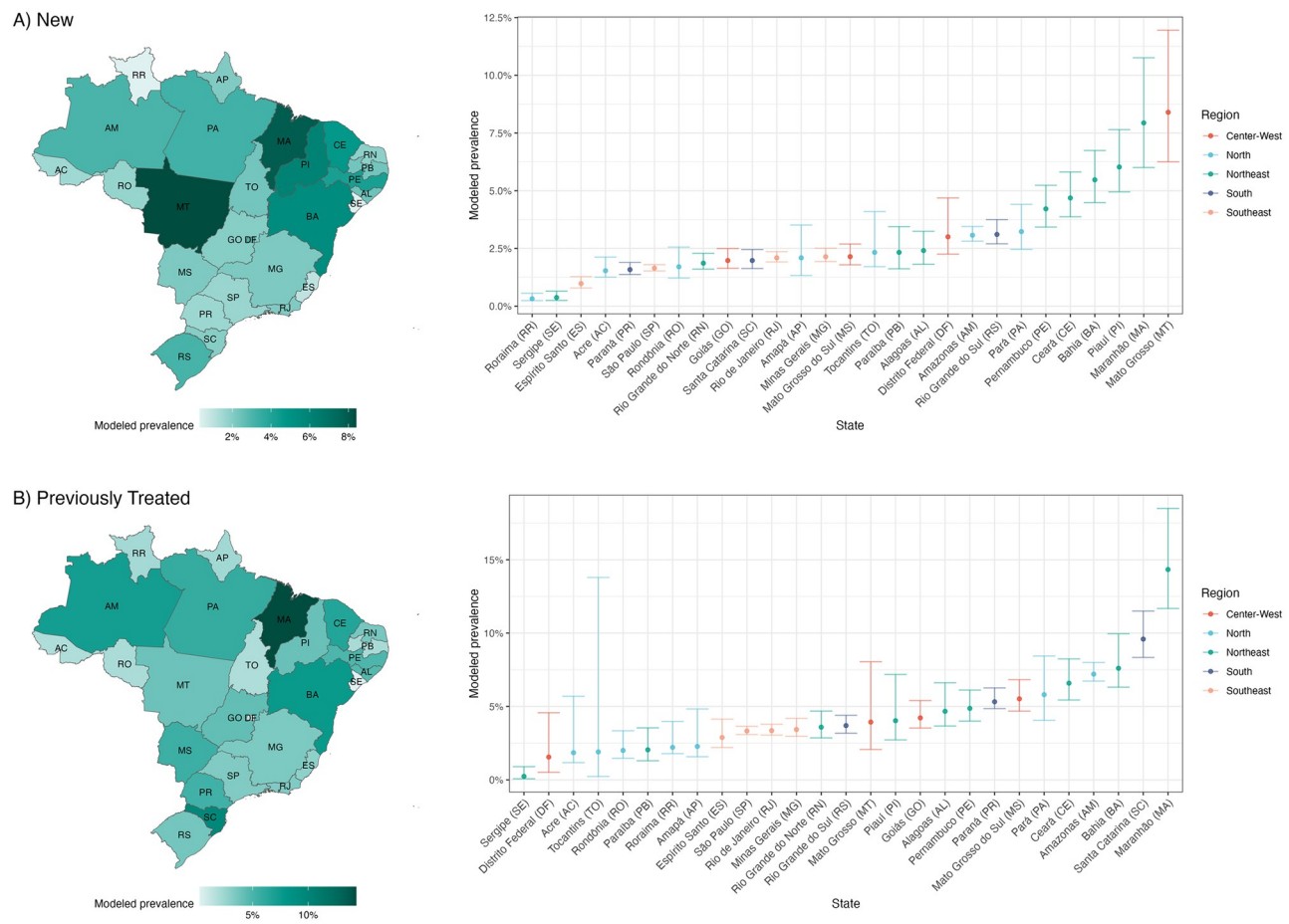

**Fig 4. Modeled levels of prevalence of rifampicin resistance by state, 2023.** Each panel presents a map of the modeled prevalence of rifampicin resistance among notified TB cases by state (left) and point estimates with 95% uncertainty intervals by each state and region (right). Two-digit codes associated with each state are listed alongside the state name on the right. Estimates have been plotted in R using the basemap shapefiles provided by the Brazilian Institute of Demography and Statistics (IBGE): https://geoftp.ibge.gov.br/organizacao_do_territorio/malhas_territoriais/malhas_municipais/municipio_2020/Brasil/BR/BR_UF_2020.zip.

2017–2023, there was heterogeneity in trends by state. Several states, such as Amazonas and Maranhão, exhibited declines in estimated resistance levels during the COVID-19 pandemic followed by increases in RR-TB notifications per 100,000 person-years through 2022.

## Comparison to WHO estimates

Our results differed from most recent estimates published by WHO's Global Tuberculosis program (Fig 3). For 2022, our point estimate of rifampicin-resistance prevalence among notified TB cases was higher for new cases (3.1% (95% UI: 3.0%, 3.3%)) compared to WHO's estimate of 1.6% (95% UI: 0.6%, 3.6%)) and lower for previously treated cases (5.1% (95% UI: 4.9%, 5.4%)) compared to WHO's estimate of 7.3% (95% UI: 1.1%, 24.0%)). For some years, our estimates fall within WHO's uncertainty intervals. Uncertainty intervals of our estimates were also narrower than those reported for WHO estimates. Our estimates of total rifampicin-resistant notifications per 100,000 person-years and incidence after accounting for under-detection were higher than WHO estimates in every year. Whereas we estimated a declining trend

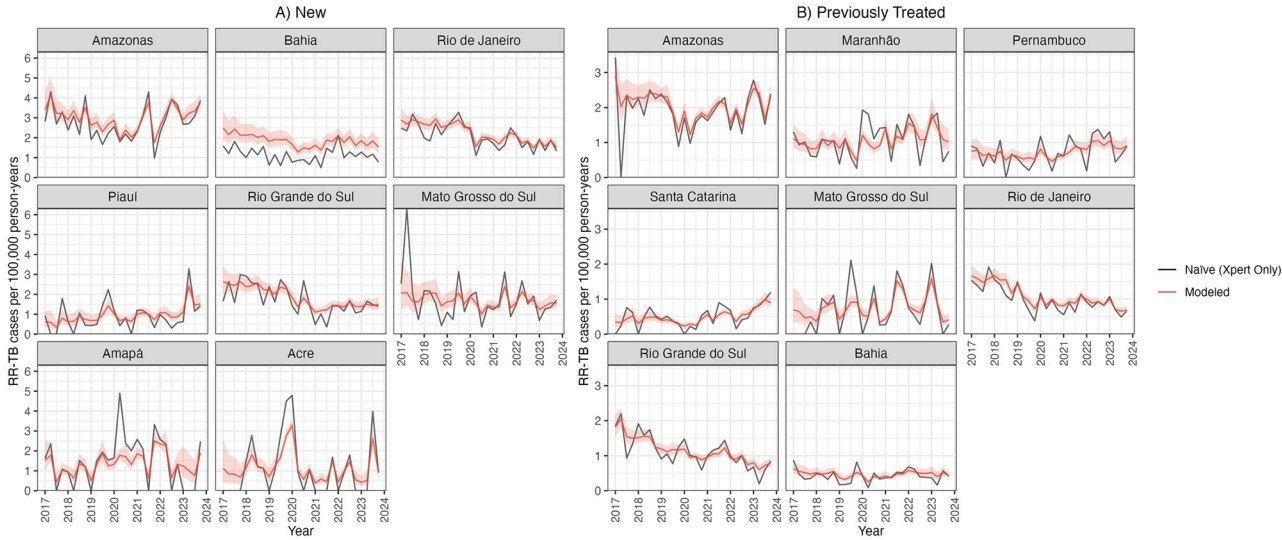

**Fig 5. Selected state-level trends in RR-TB prevalence among notified TB cases per 100,000 person-years by case type (2017–2023).** Selected states are those with the highest number of RR-TB cases among notified TB cases per 100,000 person-years and who tested at least 30% of notified TB cases in 2023. "Modeled" reflects modeled estimates and shaded 95% uncertainty intervals. "Naïve" estimates are only among Xpert MTB/RIF test results. States are ordered highest to lowest based on the number of RR-TB cases among notified TB cases per 100,000 person-years in 2023.

between 2017–2020, WHO estimated an increase over the entire period, although the uncertainty intervals for these estimates were wide.

## Alternative model specifications

Fig 6 presents comparisons between the results of the main analysis and alternative regression specifications. During the initial roll-out of Xpert in 2014, coverage was less than 3% of all notified TB cases, and average observed positivity for new and previously treated cases was 10.5% and 15%, respectively. For 2014, the model produced implausibly high estimates for both new and previously treated cases. Prevalence of rifampicin resistance was 22.0% in new cases (95% UI: 19.1%, 25.9%) and 21.3% in previously treated patients (95% UI: 17.3%, 26.8%).

The overall trend in the proportion of notified TB cases that were rifampicin resistant was robust to the inclusion of additional time invariant patient covariates. The level of prevalence of rifampicin resistance differed by 0.07 and 0.12 percentage points, on average, among new and previously treated cases, respectively. The results were also robust to the interaction of age, sex, and HIV status with time to account for any changes which patients were selected into testing over time, reflecting a 0.07 percentage point difference in new cases and a 0.01 percentage point difference in previous cases, on average.

## Discussion

We developed a novel surveillance approach to produce unbiased estimates of rifampicin resistance among notified TB cases using routinely reported RDT data. We demonstrated this approach using data from Brazil where coverage of Xpert MTB/RIF was increasing overtime, though not yet high enough to meet the threshold for continuous surveillance applied by the WHO Global TB Programme. Through a two-step approach, we modeled the risk of rifampicin resistance among individuals with recorded Xpert results and used the fitted values to impute unobserved Xpert results. After adjusting for potential biases due to systematic

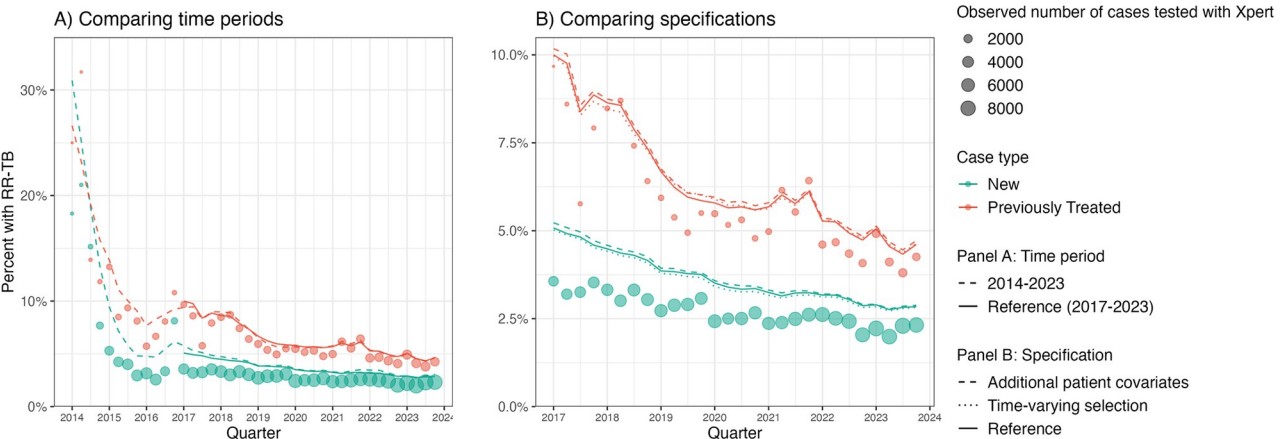

**Fig 6. Alternative model specifications.** The above compares modeled results from several alternative models by time period (A) and specification (B). (A) Extends the primary specification from 2017–2023 to include the early period of Xpert MTB/RIF implementation from 2014–2016. Across both panels, points reflect the naively estimated prevalence of rifampicin resistance—proportion of conclusive Xpert tests that are resistant. Point size indicates the number of cases with a conclusive Xpert test result. Lines reflect the modeled prevalence of rifampicin resistance among all notified TB cases for a given quarter, where line type corresponds to an alternative model. (A) Results from two alternative model specifications. The first model controls for additional patient covariates, including educational attainment, diabetes, drug consumption, tobacco consumption, alcohol consumption, whether they are experiencing homelessness, whether they are incarcerated, their immigration status, and race ("Additional Patient Covariates"). The second model fits smooth interactions between the original set of patient covariates in the model with time to determine whether there are any changes in selection over time ("Time-varying Selection"). These are compared to output from the primary specification ("Reference").

differences in Xpert access, the modeled number of rifampicin-resistant cases among TB notifications was higher than naïve estimates, especially for new cases. Modeled estimates of prevalence of rifampicin resistance among notified cases and total incidence exhibited declines between 2017–2023, and results were robust to alternative model specifications.

One would expect RDT use to be positively associated with the prevalence of resistance, resulting in naïve estimates that overestimate true resistance levels. However, our results indicate that in Brazil the naïve approach underestimated the prevalence of resistance; RDT access was higher in patient populations with a lower prevalence of resistance, specifically when looking at sex and age. While it is difficult to explore explanations for these relationships in the routine data, previous analyses have found higher probabilities of resistance among women and children in some settings, though the majority of existing evidence suggests the absence of any sex- or age-specific differences in resistance risk [19,20]. These results demonstrate the difficulty of predicting the potential direction of the bias in raw RDT resistance data, and the utility of an approach that can adjust for multiple potential sources of bias.

Our modeled estimates for Brazil between 2017–2022 differed from estimates reported by the WHO, and were more precise. There are several likely reasons for these differences. WHO uses a Bayesian hierarchical model that pools data across neighboring countries to estimate levels and trends informed by continuous surveillance data and/or nationally representative surveys of notified TB patients with bacteriologically confirmed pulmonary TB [21]. Notably, WHO estimates for Brazil rely on results from national MDR-TB/RR-TB prevalence survey from 2006–2008 and estimates the time trend using data from neighboring countries. Our model estimates both the level and time trend from national case-level testing data on a much more recent time scale (2017–2023) with a sample of over 800,000 notified TB cases, which likely contributes to narrower uncertainty intervals. Importantly, these data are already accessible to Brazil's National TB Program. Similarly, differences in estimation approach and input data likely explain the differences in the time trends estimated in our analysis compared to the

WHO's estimates–while our results reflect declining time trends in RIF resistance in the routine reporting data, the WHO estimates reflect approximately flat time trends in data from neighboring countries.

Similar approaches for estimating prevalence from data with known biases are well-developed for other surveillance topics (e.g. for HIV prevalence [22]). To our knowledge, there is no existing literature that has proposed comparable approaches for correcting potential biases in RDT data for the purposes of RR-TB surveillance. We demonstrate that our model can deliver plausible results in a setting where testing coverage is below 80%, and can be adapted to account for factors that affect selection into testing that are likely to vary across country contexts.

Our approach has several limitations. First, the approach is not theoretically guaranteed to deliver unbiased estimates of rifampicin resistance. This is only possible when the full set of factors determining testing assignment (or reasonable proxies for these factors) are known and recorded with notification data. If important factors are not available (or available but not included in the imputation model) then residual bias may remain. Although this assumption is fundamentally untestable, confidence in modelled estimates can be reinforced by a strong understanding of Xpert implementation in a given context as well as by comparing results from several alternative specifications (as demonstrated in this study), and where the opportunity exists, validation against estimates from recent national prevalence surveys. The sensitivity of model estimates to very low levels of testing coverage between 2014–2016 suggests that when coverage is low, the impact of unobserved predictors may be magnified, and therefore the approach is less reliable. Further, estimated intervals will not appropriately capture uncertainty induced by model misspecification if there are omitted or unobserved predictors. Second, our proposed approach requires analyses of line-listed notification data. Access to these data may be restricted, and so the approach we have proposed is primarily applicable to national TB programs and other agencies with access to these detailed data. This approach may not be suitable in contexts where a large fraction of TB cases are diagnosed in the private sector and thus, are not included in public sector data available to national TB programs [23]. Third, our proposed approach is not designed to investigate the causal mechanisms driving changes in RR-TB prevalence, and only reveals associations between various variables. Fourth, by design our approach estimates the prevalence of rifampicin resistances at the point of TB diagnosis, and therefore omits the acquisition of TB drug resistance by individuals receiving TB treatment. Fifth, we do not adjust our estimates for sensitivity and specificity of GeneXpert. Finally, unless otherwise indicated, the interpretation of most of our results is restricted to RR-TB prevalence among notified TB cases, as we do not observe Xpert coverage among individuals not ultimately diagnosed with TB. We attempt to correct for this by adjusting estimates for underreporting. However, this assumes that the probability of resistance is the same across notified and non-notified TB cases, which may not be valid [24]. If the probability of resistance were higher (lower) among non-notified cases, then our results would underestimate (overestimate) RR-TB prevalence in the population.

RDT data streams can deliver valuable insight on country-level RR-TB levels and trends, complementing existing WHO efforts, even when countries do not meet established thresholds for continuous surveillance nor have recent prevalence survey data available. Of the countries reporting data to the World Health Organization's Global Tuberculosis Programme, 27 countries, as of 2022, still relied on data from surveys conducted before 2015 for prevalence estimates among new cases [21]. Our method could be useful for deriving insights from existing data streams, especially where testing assignment mechanisms are known, and could potentially inform priority locations that stand to benefit the most from greater testing access as countries continue to expand coverage. Further, the applicability of this approach to other

contexts depends on having and/or establishing a strong data infrastructure, and so would be more useful with additional investments in surveillance systems. While the benefits of this method are most apparent for countries that have yet to meet established testing standards for continuous surveillance, it may also prove useful in settings where coverage is already high, given that there still may be differences in access to testing.

## Supporting information

**S1 Table. Descriptive associations between patient characteristics, selection into testing, and recorded rifampicin resistance.**
(DOCX)

## Author Contributions

**Conceptualization:** Sarah E. Baum, Ted Cohen, Nicolas A. Menzies.

**Data curation:** Daniele M. Pelissari, Fernanda Dockhorn Costa, Luiza O. Harada, Mauro Sanchez, Patricia Bartholomay.

**Formal analysis:** Sarah E. Baum.

**Methodology:** Sarah E. Baum, Daniele M. Pelissari, Luiza O. Harada, Mauro Sanchez, Patricia Bartholomay, Ted Cohen, Marcia C. Castro, Nicolas A. Menzies.

**Supervision:** Marcia C. Castro, Nicolas A. Menzies.

**Visualization:** Sarah E. Baum.

**Writing – original draft:** Sarah E. Baum.

**Writing – review & editing:** Sarah E. Baum, Daniele M. Pelissari, Fernanda Dockhorn Costa, Luiza O. Harada, Mauro Sanchez, Patricia Bartholomay, Ted Cohen, Marcia C. Castro, Nicolas A. Menzies.

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
