## [Decision Letter · Decision Letter 0]

17 Oct 2024

Dear Ms. Baum,

Thank you very much for submitting your manuscript "Surveillance for TB drug resistance using routine rapid diagnostic testing data: Methodological development and application in Brazil" for consideration at PLOS Computational Biology.

As with all papers reviewed by the journal, your manuscript was reviewed by members of the editorial board and by several independent reviewers. In light of the reviews (below this email), we would like to invite the resubmission of a significantly-revised version that takes into account the reviewers' comments.

The reviewers offer valuable complementary insights from both methodological and epidemiological perspectives, providing essential guidance for enhancing the manuscript. Please also ensure that you are in compliance with our code-sharing policy.

We cannot make any decision about publication until we have seen the revised manuscript and your response to the reviewers' comments. Your revised manuscript is also likely to be sent to reviewers for further evaluation.

Sincerely,

Claudio José Struchiner, M.D., Sc.D.

Academic Editor

PLOS Computational Biology

Virginia Pitzer

Section Editor

PLOS Computational Biology

The reviewers offer valuable complementary insights from both methodological and epidemiological perspectives, providing essential guidance for enhancing the manuscript.

Reviewer's Responses to Questions

**Comments to the Authors:**

Reviewer #1: Thank you for the opportunity of reviewing this interesting study about a key issue around TB. Drug resistance is a complex and serious problematic and being able to access accurate estimations is important for both research and local policy-making. This work has merit and I recommend its publication if the authors are able to address the points below.

Abstract

Lines 7-8: “RDT data collected in the course of routine (but not universal) use may provide biased estimates of resistance”

Unclear what the authors mean by ‘not universal’ in this context. Do they mean that RDT data are not collected in all countries? Or that RTD testing is not offered to all TB patients even within the same country? Please clarify and explain why the estimate may be biased.

Lines 17-18: “Our estimates of RR-TB incidence were considerably more precise than WHO estimates for the same time period”

This is vague. The authors should clarify why they believe their estimates more precise, according to which metric.

Author Summary

Lines 30-32: “Applying this approach to Brazil, we find that modeled estimates were higher than naïve estimates, and were more precise compared to estimates produced by the World Health Organization.”

Same point as in the abstract, please clarify how these results are more precise than WHO’s estimates.

Introduction

Lines 56-57: “However, these data can yield biased estimates if RDT coverage is not universal, or there are systematic differences in access to testing.”

The authors should clarify what they mean by systematic differences in access to testing. Something systematic should be easier to estimate or predict, maybe they meant systemic differences? It would be good if they could provide examples.

Lines 70-72: “countries that rely exclusively on national surveys to estimate prevalence among new cases accounted for almost 25.7% of the estimated annual number of incident TB cases”

Suggest rephrasing with the following for clarity: “countries that rely exclusively on national surveys to estimate drug resistance prevalence among new cases accounted for almost 25.7% of the estimated annual number of global incident TB cases”

Method

Lines 170-184: This whole paragraph is very wordy. I appreciate the necessity to have a step-by-step explanation on how the statistical analysis is performed, but this should come with either formulas or a schematic, to make this visual and easier for the reader to follow. At the same time, the authors should go more into details on how WHO’s estimations are performed and how that method differs from their own, comparing the formulas/schematics.

Results

Lines 318-327: (1) The authors explain how their results differ from the WHO ones, however they fail to mention that estimations for latest years fit into WHO’s confidence intervals, which should be highlighted. (2) The authors argue that WHO’s confidence intervals are wide, however I would argue that only the ones for previously treated cases are somewhat wide, while in new and total cases these are narrow, with the modelled uncertainty intervals being extremely narrow. The authors should address why the uncertainty in their estimations is so small. (3) Results also show a decreasing modelled trend compared to a somewhat constant trend in WHO estimations, however this difference seems to decrease as the model runs in recent years in new and total cases, while it seems to keep decreasing in previously treated. It would be good if the authors could expand on this either here or in the discussion.

Discussion

The authors mention how WHO’s estimates for Brazil rely on results from national MDR-TB/RR-TB prevalence survey from 2006-2008 and estimate the time trend using data from neighbouring countries. While their model estimates both the level and time trend from case-level testing data from Brazil’s National TB Program on a much more recent time scale (2017-2023). This is the reason why they are justified in arguing that their results might be more precise and it should be mentioned much earlier in the paper including the abstract, even just a few words about this method using recent local data and leave the lengthier explanation to the discussion.

Additionally, I have not seen any mention of the test accuracy being taken into account, can the authors please clarify whether this was the case and, if not, mention this in the limitations and discuss if and how their results might change if they did.

Reviewer #2: Dr Baum and colleagues present in their manuscript a novel method for estimating/correcting TB resistance trends by using a hierarchical generalized additive regression model applied to existing rapid diagnostics tests results. The manuscript is exceptionally clear in their explanation of the method, and I commend the authors for their clarity and transparency in the description of methods as well as recognition of limitations.

In my capacity of infectious disease epidemiologist, I will limit my comments to the impact of this method for the field and the potential limitations from the epidemiological point of view, and will leave the statistical details for other experts in these methods.

First I need to say that I did not find major problems with this manuscript, so my suggestions are only minor.

The method proposed by Baum and colleagues is a robust approach that leverages from existing (if imperfect) data on RDTs. The method is very relevant to the field of TB burden estimation at local and global level and as I read the paper I could see how this could have a large impact on how we understand the real burden of TB resistance globally. A couple of considerations and suggestions:

- Under-reporting correction assumptions: The assumption behind under reporting corrections is a very important one. Although we can't prove the contrary, it is mostly likely that the probability of resistance is not the same between notified and non-notified TB cases. Although the authors recognise how this can be a potential issue, it would be useful to understand how this could be tackled in the future and what are the implications for the current results.

- Applicability in other contexts: The case of Brazil is very particular, as they have an under the threshold fraction of RDT use at point of diagnosis, but they have a large, rather complete surveillance system (SINAN, SISCEL etc) which allows for a statistical model like the one presented to be designed and analysed. The authors do recognise this in their discussion but I believe that one of the aims of this work beyond this publication should be to propose a path to better estimates of RR-TB trends from those made by WHO. It would be useful if the authors could more accurately point to the settings in which this method could actively help improve estimates and propose a path forward.

**Have the authors made all data and (if applicable) computational code underlying the findings in their manuscript fully available?**

Reviewer #1: Yes

Reviewer #2: **No: **As I mentioned, the authors make their code available upon request, but in reality, and more from this journal, one expect to have easy access to code and scripts behind the methods, understanding that some data cannot be shared.

PLOS authors have the option to publish the peer review history of their article (what does this mean?). If published, this will include your full peer review and any attached files.

Reviewer #1: No

Reviewer #2: No
---

## [Decision Letter · Decision Letter 1]

20 Nov 2024

Dear Ms. Baum,

We are pleased to inform you that your manuscript 'Surveillance for TB drug resistance using routine rapid diagnostic testing data: Methodological development and application in Brazil' has been provisionally accepted for publication in PLOS Computational Biology.

Best regards,

Claudio José Struchiner, M.D., Sc.D.

Academic Editor

PLOS Computational Biology

Virginia Pitzer

Section Editor

PLOS Computational Biology

Feilim Mac Gabhann

Editor-in-Chief

PLOS Computational Biology

Jason Papin

Editor-in-Chief

PLOS Computational Biology

Reviewer's Responses to Questions

**Comments to the Authors:**

Reviewer #1: My comments have been fully addressed thus I am satisfied with the current state of the manuscript.

Reviewer #2: I'm satisfied with the authors' response to my comments and the changes made to the manuscript. I have no further comments

**Have the authors made all data and (if applicable) computational code underlying the findings in their manuscript fully available?**

Reviewer #1: None

Reviewer #2: Yes

PLOS authors have the option to publish the peer review history of their article (what does this mean?). If published, this will include your full peer review and any attached files.

Reviewer #1: No

Reviewer #2: No

---

## [Editor Report · Acceptance letter]

5 Dec 2024

PCOMPBIOL-D-24-01238R1 

Surveillance for TB drug resistance using routine rapid diagnostic testing data: Methodological development and application in Brazil

Dear Dr Baum,

I am pleased to inform you that your manuscript has been formally accepted for publication in PLOS Computational Biology. Your manuscript is now with our production department and you will be notified of the publication date in due course.

With kind regards,

Anita Estes
